# The Immune–Genomics of Cholangiocarcinoma: A Biological Footprint to Develop Novel Immunotherapies

**DOI:** 10.3390/cancers17020272

**Published:** 2025-01-15

**Authors:** Antonella Cammarota, Rita Balsano, Tiziana Pressiani, Silvia Bozzarelli, Lorenza Rimassa, Ana Lleo

**Affiliations:** 1Hepatobiliary Immunopathology Laboratory, IRCCS Humanitas Research Hospital, 20089 Rozzano, Italy; 2Department of Biomedical Sciences, Humanitas University, 20090 Pieve Emanuele, Italy; rita.balsano@humanitas.it (R.B.); lorenza.rimassa@hunimed.eu (L.R.); 3Medical Oncology and Hematology Unit, Humanitas Cancer Center, IRCCS Humanitas Research Hospital, 20089 Rozzano, Italy; tiziana.pressiani@humanitas.it (T.P.); silvia.bozzarelli@humanitas.it (S.B.); 4Division of Internal Medicine and Hepatology, Department of Gastroenterology, IRCCS Humanitas Research Hospital, 20089 Rozzano, Italy

**Keywords:** cholangiocarcinoma, cancer-associated fibroblasts, tumor-immune microenvironment, immunotherapy, immune targets

## Abstract

Cholangiocarcinoma (CCA), a rare but aggressive bile duct cancer, is on the rise globally, especially a type called intrahepatic CCA. Recent research has revealed that CCA is highly diverse at the genetic and immune levels, and creates an environment that weakens the immune system, making it harder for the body to fight the cancer. A big step forward has been combining chemotherapy with immune checkpoint inhibitors (ICIs), a type of immunotherapy, which is now the standard for advanced CCA. Although these treatments extend survival, the improvement is still limited. Thus, scientists are exploring new strategies, like targeting multiple immune pathways and focusing on specific genetic vulnerabilities to boost treatment response. This work summarizes advancements in our understanding of CCA at the molecular level, which have enabled the development of personalized therapies, and highlights ongoing efforts to make CCA a more treatable disease through innovative and targeted approaches.

## 1. Introduction

Cholangiocarcinoma (CCA) accounts for approximately 3% of all gastrointestinal cancers and belongs to a heterogeneous group of aggressive cancers arising from the epithelial component of the biliary tree system [1]. It is classified according to the anatomical location of origin as intrahepatic (iCCA) or extrahepatic (eCCA), including perihilar (pCCA) and distal tumors (dCCA), gallbladder cancer (GBC), and ampullary cancers. Overall considered a rare tumor entity, CCA incidence has been rising over the past decades in most countries, mainly due to an increased diagnosis of iCCA [2]. Risk factors vary worldwide and encompass viral (hepatitis B and C) and non-viral (primary sclerosing cholangitis) chronic liver disease, coledocolithiasis, structural abnormalities of the biliary tree, and, especially in Asian countries, liver flukes (Opisthorchis viverrini and Clonorchis sinensis). Rising trends in Western countries have been linked to the increase in metabolic syndrome and its associated liver chronic diseases (metabolic dysfunction-associated steatotic liver disease [MASLD] and steatohepatitis [MASH]) [3]. However, causative links with metabolic syndrome remain elusive, and, in many cases, CCA arises on a seemingly “healthy” liver [4]. The lack of identifiable risk factors in a large proportion of patients has limited the implementation of dedicated early detection programs, affecting the prognosis of CCA, which remains dismal due to late-stage diagnosis and resistance to conventional therapies [1,2].

Translational research has shown that CCA is a molecularly heterogeneous disease in which the tumor-immune microenvironment (TIME) plays an active role in the pathogenesis and progression [5]. The TIME of CCAs hosts a variety of immune cells, endothelial cells, and stromal components, many of which promote immune-suppressive programs [6,7,8,9,10,11,12,13]. Cancer-associated fibroblasts (CAFs) are a key population within the TIME, contributing to the immunosuppressive landscape and correlating with worse survival [12,13,14].

Advances in the study of CCA tumor biology have been a major driver toward the implementation of novel treatments. First-line combinations with anti-programmed death-(ligand)-1 (PD-(L)1) antibodies plus cisplatin and gemcitabine have set a new standard of care in the management of patients with unresectable or metastatic CCA [15,16]. Moreover, the identification of druggable genetic alterations in nearly half of CCAs has established a paradigm change, offering pathways for personalized therapies in a sizeable proportion of patients [17,18,19,20,21,22]. However, benefits from systemic treatments remain overall modest spurring interest in exploiting novel immune molecules as therapeutic targets.

This review aims to provide a detailed overview of the immune biology of CCAs and explores its therapeutic implications in the context of both established and emerging treatment strategies.

## 2. Biology of CCA

### 2.1. Tumor Phenotypes

The genomic landscape of CCA is highly heterogeneous. A study by the Cancer Genome Atlas (TCGA) revealed that the disease has an intermediate mutation burden with overlapping, low-penetrance driver mutations in several signaling pathways [23]. This integrated analysis of somatic mutations, RNA expression, copy number aberrations (CNAs), and DNA methylation distinguished four molecular subclasses leveraging a small cohort of viral-negative and fluke-negative CCAs. Cluster 1, which included mostly eCCA, comprised molecularly atypical tumors characterized by hypomethylation and lack of CNAs or recurrent driver mutations previously described in CCA. Cluster 2 was enriched in genomically unstable tumors with multiple chromosomal deletions. Of note, isocitrate dehydrogenase (*IDH*) hotspot mutations were only found in this group. Other features were high representation of mitochondrial genes and low expression of chromatin modifier genes, pointing to a possible regulatory role of these pathways by (R)-2-hydroxyglutarate (2HG), the oncometabolite that accumulates in *IDH1*-mutated CCAs. Cluster 3 contained CCA-common CNAs (e.g., 1p loss and 1q gain) and was enriched in fibroblast growth factor receptors 2 (*FGFR2*) alterations, while cluster 4 had high-level amplification of Cyclin D1 (*CCND1*) and the highest methylation profile.

Differences in the anatomic site of origin and etiologies further contribute to the genomic heterogeneity seen in CCAs [24,25,26]. A recent analysis conducted on 412 patients with CCAs from Italy and Japan showed that the majority of iCCAs had mutations in epigenetic genes, whereas eCCA and gallbladder tumors had mutations in *TP53* and cell cycle genes [27]. Accordingly, more than half of the cases in a cohort of Western patients with idiopathic pCCA harbored *TP53* alterations, along with other oncogenic signaling pathways encompassing Rat sarcoma-mitogen-activated protein kinase (*RAS/MAPK*) (59%), phosphatidylinositol-3-kinase-mammalian target of rapamycin (*PI3K/mTOR*) (29%), and neurogenic locus notch homolog protein (*NOTCH*) (14%) [28]. When related to liver flukes or viral hepatitis, CCAs showed unique mutation and DNA hypermethylation profiles which included a higher incidence of *TP53* mutations and lower rates of *IDH* mutations [29]. These findings are in keeping with the low prevalence of *IDH* (5%), Protein polybromo-1 (*PBRM1*) (1%), and BRCA1-associated protein-1 (*BAP1*) (1%) mutations reported in a Chinese iCCA cohort [30].

Tumor profiling has allowed the definition of distinct molecular phenotypes of CCA with implications for patient prognosis and response to therapy [31,32,33]. Based on gene expression, single-nucleotide polymorphisms, and mutational profiles, 149 iCCA were categorized into an inflammation or proliferation class depending on whether or not they had predominantly activated inflammatory (e.g., Signal Transducer and Activator of Transcription 3 [*STAT3*]) or oncogenic (*RAS/MAPK* and mesenchymal-epithelial transition factor [*MET*]) signaling pathways, respectively [32]. Of note, the inflammation class had a more indolent clinical course. Combined genomic and transcriptomic analyses have also introduced similar classifications in eCCA. A study of 189 eCCA revealed four distinct clusters: mesenchymal (47%), proliferation (23%), metabolic (19%), and immune (11%) [33]. The mesenchymal cluster, which was associated with poor prognosis, was characterized by increased epithelial-to-mesenchymal transition (*EMT*), activation of tumor growth factor beta (*TGF-β*) signaling, and an extensive desmoplastic reaction. The proliferation cluster harbored over-represented cell cycle and DNA repair pathways, *HER2* alterations, and *MYC* targets, and was enriched in dCCA. The metabolic cluster presented with an *HNF4A*-driven hepatocyte-like phenotype, deregulated pathways involved in the metabolism of bile acids, fatty acids, and xenobiotics, and infiltration of gamma-delta T cells. Lastly, the immune cluster had high levels of lymphocyte infiltration and PD-1/PD-L1 expression.

While these biologically defined classes of CCA have been an elegant attempt to tackle disease heterogeneity, their clinical use has remained limited. However, these studies had the value of showing that CCAs are studded with potential therapeutic vulnerabilities. Tumor profiling in a clinically relevant window has reported druggable genetic alterations in 40–60% of the patients with CCAs, particularly among iCCA [17,34,35]. These include *FGFR2* rearrangements or fusions (7–15%) and *IDH1* mutations (10–20%), which are found almost exclusively in iCCA. Less frequent targetable alterations are B-Raf proto-oncogene serine/threonine kinase (*BRAF*) V600E (5%), *RET* (1%), Neurotrophic tyrosine receptor kinase (*NTRK*) (1%), and microsatellite instability (*MSI-H*) (1%). Gallbladder cancers and eCCA are more target-devoid, with *HER2* aberrations (15–20% amplifications/overexpression and 1–2% mutations) being the most frequently reported. These molecular subsets seem to harbor a unique genomic landscape. Tumors harboring *FGFR2* alterations have been associated with *BAP1* co-mutations, while *IDH1*-mutant tumors exhibit downregulation of chromatin remodeling genes and an *IDH*-specific hypermethylation profile, which is interestingly maintained across multiple histologies [23,34]. Lastly, these molecular subtypes may be correlated with distinct survival outcomes, with *FGFR2*-altered tumors showing improved prognosis in clinical cohort studies [36,37].

### 2.2. Tumor-Immune Microenvironment in CCA

The TIME has an integral role in tumor growth and progression in CCA as in other solid tumors [5]. CCAs are largely characterized by an immune-suppressive landscape resulting from the predominance of regulatory or dysfunctional immune populations, reduced infiltration by cytotoxic immune cells, and extensive desmoplasia. Immune profiling studies have described an intricate crosstalk between tumor and different TIME cell compartments, including myeloid-derived suppressor cells (MDSCs), tumor-associated macrophages (TAMs) and neutrophils (TANs), regulatory T cells (T regs), and CAFs that sustain these immune-suppressive programs in CCA [6,7,8,9,10,11,12,13]. This paragraph focuses on the innate and adaptive immune populations composing the TIME of CCAs, while CAFs are treated in greater detail in Section 2.3.

MDSCs and TAMs are abundant pro-tumorigenic and immune-suppressive populations in the TIME of CCAs [38,39]. MDSCs are organized into two major subsets of monocytic (M-MDSC) and polymorphonuclear (PMN) MDSCs, whereas TAMs have mostly a *STAT3*-mediated M2 polarization (CD163+) [40,41,42]. Recently, TANs have also been demonstrated to be a functionally relevant TIME cell type, with active contributions to cancer growth and immune escape in CCA [8,9]. Mostly derived from peripheral blood neutrophils, TANs are a highly plastic population recruited in the liver following chemokine release by tumor and resident immune and stromal cells. In CCAs, the abundance of all these immune-suppressive cell types has been correlated with tumorigenesis, disease progression, increased infiltration and differentiation of regulatory cells, and ultimately worse clinical outcomes [43,44,45,46]. However, the contribution and interdependency of each cell population in creating the poorly immunogenic TIME of CCAs remain largely unknown, posing major challenges to the implementation of therapeutic strategies that could overcome resistance to first-generation immunotherapies. These populations establish a series of delicately intertwined communications in the TIME of CCAs. For instance, TAMs and TANs were proven to have additive pro-tumor effects in vitro, and depletion of TAMs in mice did not improve survival due to a compensatory increase in MDSCs, requiring dual blockade of TAMs and MDSCs to obtain immune-sensitive CCA preclinical models [6,9].

The TIME of CCAs is also enriched with T regs supporting the immune-suppressive programs that are pathognomonic of the disease [10]. Under physiological conditions, the transcriptional factor forkhead box protein P3 (*FOXP3*) determines the specific functional phenotype of T regs [47,48]. These regulatory immune cells restrict the activity of antigen-presenting, natural killer (NK), and CD8+ cytotoxic T cells via multiple mechanisms. These include the release of anti-inflammatory molecules, such as interleukin (IL)-10, IL-35, and TGF-β1, the consumption of IL-2, and the expression of immune checkpoints. In CCA, it has been suggested that multiple oncogenic pathways, such as *PI3K/Akt* and Dickkopf-1 (*DKK1*)-*Wnt*, have pivotal roles in the recruitment and differentiation of T regs [49,50]. Further, a sc-RNA-seq study identified a phenotype of intratumoral T regs with enhanced immune-suppressive roles resulting from alterations in the network of transcription factors between tumor-infiltrating and peritumoral T cells [10].

Lastly, the spatial distribution of these populations further contributes to the TIME polarization and retains prognostic significance [51,52,53,54,55,56]. Immunohistochemical analyses of immune markers on 74 iCCA resected samples showed that T cell and immune checkpoint markers are mainly enriched at the tumor margins in CCA [57]. Here, their relative abundance and spatial organization exert distinct prognostic effects. In detail, tumor-infiltrating CD4+ or CD8+ T cells, along with a high density of B cells, particularly when clustered into tertiary lymphoid structures (TLSs), have been linked with more favorable survival, whereas FOXP3+ T reg cells and CD163+ macrophages with poorer patient outcomes [53,58,59,60]. By contrast, the clinical relevance of the quantification of regulatory immune checkpoints in CCA remains controversial. For instance, PD-L1, a renowned biomarker of response to immunotherapy in other solid tumors, has been reported in 9–70% of the cases in CCA but its association with patient-relevant outcomes remains unclear [61,62,63,64,65]. Figure 1 provides an overview of the main cell types composing the TIME of CCAs and their respective functional roles.

### 2.3. Cancer-Associated Fibroblasts in CCA

In desmoplastic tumors like CCA, CAFs are a well-represented population in the TIME. In iCCA, CAFs have been shown to originate mainly from hepatic stellate cells (HSCs) and portal fibroblasts, whereas their origin is less clear in eCCA [13,66]. Upon malignant transformation, these cells activate, proliferate, produce extracellular matrix (ECM), and assume a pro-inflammatory secretory profile [13,67]. As in other tumors, CAFs in CCAs have largely been identified by the expression of alpha-smooth muscle actin (α-SMA). However, collagen type I alpha-1 (COL1A1), vimentin, fibroblast activation protein (FAP), platelet-derived growth factor receptor-alpha (PDGFR-α), and platelet-derived growth factor receptor-beta (PDGFR-β) are also commonly expressed markers [68]. In recent years, studies with single-cell resolution have enabled the identification of a pan-CAF signature that recapitulates the different shades of CAFs in CCAs. Patients whose tumors have abundant CAFs have been found to have a worse prognosis [13,69].

CAFs have broadly recognized tumor-promoting roles in solid tumors, including CCAs, whereas their potential tumor-suppressive activity remains to be better defined. High-resolution studies have described multifaceted interactions of CAFs with the tumor, immune, and endothelial compartments of CCAs resulting in the activation of multiple pathways involved in tumor stiffness, tumorigenesis, immune suppression, and neo-angiogenesis [12,13].

A pivotal study in mice with validation in human cohorts revealed that CCAs have a heterogeneous and highly plastic population of CAFs [13]. The two predominant subtypes have been defined as myofibroblastic CAFs (myCAFs) due to the deposition of matrix components and inflammatory CAFs (iCAFs) due to the secretion of cytokines and growth factors. Although different analyses have adopted different classifications, there is an overall agreement about these two subtypes. Other clusters identified in CCAs include antigen-presenting CAFs (apCAFs) enriched in major histocompatibility complex II, vascular CAFs (vCAFs) involved in tumor neo-angiogenesis, epithelial to mesenchymal-like CAFs (eCAFs) expressing EMT markers, and lipofibroblasts (lCAFs) implicated in the regulation of lipid metabolism [12].

The deposition of ECM components is one of the most studied roles of CAFs and myCAFs are the key subpopulation with this function [12,13]. In CCAs, ECM proteins promote tumorigenesis and invasion via self-sustaining stroma-tumor interactions as increased tumor stiffness activates CAFs in return. The expression of many of these ECM components, such as laminin, osteopontin, and several matrix metalloproteinases (MMPs), has been associated with more advanced tumors and poor patient prognosis [14,70,71]. Notably, the fibrillar collagen COL1A1, which is one of the most abundant ECM proteins secreted during tumorigenesis, cooperates to increase tumor stiffness, alter TIME organization, and enhance MMP activity [13]. However, despite beliefs that COL1A1 would also contribute to tumor progression, this has not been confirmed in iCCA where this role has been reported for hyaluronic acid but not COL1A1.

Another recognized function of CAFs is the release of growth factors implicated in cancer growth and progression. For example, myCAFs produce heparin-binding epithelial growth factor (HB-EGF) which induces *EGFR* activation in CCA cells, ultimately promoting migration and invasiveness [72]. In return, on HB-EGF stimulation, CCA cells produce TGF-β1, which stimulates HB-EGF expression in myCAF, resulting in a constant positive feedback loop sustaining CCA progression. Moreover, the *EGFR* and *TGF-β1* signaling pathways induce EMT transition and immune suppression in CCAs [73]. Another pro-tumorigenic growth factor produced by CAFs is the hepatocyte growth factor (HGF), which binds to c-MET on tumor cells and activates the downstream MAPK, extracellular signal-related kinase (*ERK*), and *PI3K* pathways [13,74]. PDGF-β, which activates the Hedgehog signaling (Hh) pathway, has a similar role [74,75]. CAFs also release chemokines and cytokines, such as C-X-C motif chemokine 12 (CXCL12) and interleukin (IL)-1β, that promote CCA proliferation, migration, and invasion through *ERK1/2* and *PI3K/Akt* signaling cascades [76,77,78]. Notably, high expression of the CXCL12 binding receptor CXCR4 was correlated with features of aggressive disease and shorter survivals in patients with iCCA [79].

Furthermore, CAFs have immunomodulatory functions and emergent roles in resistance to immunotherapy in solid tumors, including CCAs [68,80,81,82]. CAF-secreted prostaglandin E2 (PGE2) and TGF-β contribute to the establishment of an immune-suppressive niche populated by abundant and activated T regs and scarce and dysfunctional effector cells [68]. For instance, myCAFs, which have been reported to have the highest expression of TGF-β ligands, were associated with a signature of resistance to immunotherapy (IPRES) in CCAs [82]. Through the release of IL-13, IL-6, CXCL12, and macrophage colony-stimulating factor, CAFs have also been shown to recruit the immune-suppressive populations of MDSCs and TAMs, and support M2 polarization [83,84]. The activation of STAT3 and the production of chemokine ligand 2 (CCL2) following FAP expression by CAFs enhances the recruitment of MDSCs. Further, the ECM compounds secreted by CAFs create a physical barrier to immune cell recruitment and may further favor the selection of immune-suppressive populations [85]. In pancreatic cancer, MMPs are chemotactic for leukocytes and COL1A1 increases MDSC accumulation and suppression of CD8+ T cells; this remains yet to be confirmed in CCA [86].

Additionally, CAFs contribute to tumor vascular and lymphatic neo-angiogenesis [87]. In CCAs, CAFs are a vast source of vascular endothelial growth factor (VEGF) in the tumor stroma. A sc-RNA-seq study highlighted a specific CD146+ vCAF subpopulation that is highly abundant in CCAs and fosters neo-angiogenesis through the activation of the IL-6/IL-6R axis among other pathways [12]. Notably, HSCs can also secrete pro-angiogenic factors, including PDGF, FGF, EGF, and angiopoietin-1 and -2, but their role in CCA neo-angiogenesis remains to be further explored [88].

Preliminary evidence has suggested CAF involvement in other tumor-promoting roles, including metabolic reprogramming, senescence, and dysfunctional protein post-translational modifications, which remain a matter of study in CCAs [68].

In summary, CAFs exert pro-tumorigenic and pro-inflammatory functions in the TIME of CCAs through a series of complementary mechanisms. While these functions may be preferentially executed by different CAF subpopulations, the existence of clusters with mixed features also suggests that CAF subpopulations are not strictly separate. In studies conducted in pancreatic tumors, the Janus Kinase/Signal Transducer and Activator of Transcription (JAK/STAT) signaling pathway may be involved in myCAF-iCAF plasticity programs [89]. Thus, CAFs possibly exist in a continuum of states adapting to tumor and TIME changes. However, as functional studies remain limited, the exact mechanisms driving these dynamic changes in CCA deserve further investigation. Figure 2 summarizes the main roles of CAFs in CCA.

### 2.4. Immune-Based Classifications of CCA

Immune checkpoint blockade has transformed cancer therapy, but its efficacy in unselected populations, including CCAs, remains limited [15,16]. Sensitivity to anti-PD-(L)1 antibodies is largely dependent on the pre-existing cellular makeup and functional state of the TIME. Thus, the advent of high-resolution techniques has provided an unprecedented opportunity to identify TIME phenotypes that predict responsiveness or resistance to these therapies [90].

Foundational studies of the immune infiltrate contexture in solid tumors are mainly based on low-to-moderate resolution analyses (e.g., bulk sequencing and immunostainings). These studies divide the TIME into three main classes, namely infiltrated–excluded (i.e., poorly immunogenic or immunologically “cold”), infiltrated–inflamed, and infiltrated–TLS [91]. These classes, which are likely to evolve along a spectrum of sub-statuses, are characterized by the organization of immune cells at the invasive tumor margin, inside the tumor area, or in aggregates that organize similarly to lymph nodes, respectively. Despite TIME classes assuming various names across different studies, their main composition and behavior are maintained, making this broad classification handy for interpreting existing evidence in CCA.

Using transcriptomics from a large cohort of iCCAs, a study identified four immune subclasses associated with distinct immune escape mechanisms and patient outcomes: immune-desert, myeloid, mesenchymal, and immunogenomic [92]. The immune-desert subset, characterized by low expression of TIME gene signatures, was the main subclass (45%) supporting the notion that CCA is an immunologically “cold” tumor. The myeloid and mesenchymal clusters were enriched with monocyte-derived and fibroblast gene signatures, respectively. Lastly, the immunogenomic cluster was the least represented (about 11% of CCAs). It featured infiltration by innate and adaptive immune cells with highly activated inflammatory and immune checkpoint pathways and was associated with prolonged survival.

Another recent study proposed a novel Stroma, Tumor and Immune Microenvironment (STIM) classification of iCCAs from the analysis of 122 samples with further validation in five independent datasets [82]. This classification identified five immune classes of iCCA, overall split between non-inflamed (65%) and inflamed (35%) tumors. The non-inflamed tumors comprised three major classes: a “desert-like” class (20%) with scarce immune cells but enriched in T regs, a “hepatic stem-like” class (35%) rich in M2 macrophages and tumors with *IDH*, *BAP1*, or *FGFR2* alterations, and a “tumor classical” class (10%) characterized by the activation of cell cycle pathways and worse patient survivals. The inflamed tumors were further divided into an “immune classical” (10%) group, which was infiltrated by adaptive immune cells like CD8+ T cells, and an “inflammatory stroma” (25%) group, which presented extensive desmoplastic reaction, T cell exhaustion, and *KRAS* mutations.

Further, a comprehensive analysis using whole-exome, RNA, and T cell receptor (TCR) sequencing together with multiplexed immunofluorescence clustered iCCAs into sparsely, heterogeneously, and highly infiltrated subgroups with distinct immunogenomic features [93]. Sparsely infiltrated tumors had active copy-number loss of clonal neoantigens and heterogeneous immune infiltration which played an important role in the subclonal evolution across tumor regions. Highly infiltrated tumors presented with wide immune activation and a homogeneous TCR repertoire across tumor regions. However, these features were offset by T cell exhaustion and impaired antigen presentation. Interestingly, FGFR2 mutations and fusions correlated with low mutation burden and reduced immune infiltration. In keeping with this, other recent analyses exploring immune biomarkers in molecularly defined subgroups found low PD-L1 expression in the *FGFR2*-altered cluster [94,95].

Lastly, a Chinese study provided a four-tiered immune classification of iCCAs based on the spatial distribution and abundance of TLSs [96]. Overall, tumors sat on a continuum with edges represented by class I (28%) iCCAs with TLSs located in the peri-tumoral region and class IV (12%) iCCAs with TLSs mainly found in the tumor region. In between, classes II (31%) and III (28%) had heterogeneous distribution patterns. Of note, the maturation stage of TLSs also varied depending on their localization. Intra-tumoral TLSs had features of more mature lymphoid structures with germinal centers and more intact tumor–host immune response, and were associated with improved survival.

In summary, the integration of genomic, transcriptomic, and immunological data has enabled the identification of distinct patterns describing the TIME of CCAs, particularly in iCCAs. While there is no current consensus favoring one classification system over another, these approaches collectively highlight that CCAs predominantly harbor an immunosuppressive TIME. However, even though these tools may be helpful in stratifying patients into immunologically and prognostically different subtypes, their confinement to resected iCCAs and lack of prospective validation limit their implementation in routine patient care.

## 3. Targeting the TIME of Clinics

Immunotherapy leverages the intrinsic ability of the immune system to recognize and eliminate “non-self” entities, including tumors [97]. Having revolutionized the treatment of many cancers, immunotherapy also warranted investigation in CCA, where standard-of-care first-line chemotherapy with cisplatin and gemcitabine resulted in a dismal median OS below one year [98]. In keeping with the finding that CCA largely lacks cytotoxic immune infiltration, single-agent immune checkpoint inhibitors (ICIs), which rely on pre-existent anti-tumor immunity, showed a limited activity (overall response rate [ORR] 5.8–13%) for advanced CCA [99]. However, the efficacy improved by combining ICIs with chemotherapy, an approach used to establish a more proficient anti-tumor immune infiltrate. In fact, although cisplatin and gemcitabine, the cornerstone treatment for unresectable CCA, have immune-suppressive effects on peripheral blood cell counts, their cytotoxic activity in the tumor bed has been proven to be immunogenic, fostering antigen presentation, depleting the TIME of immune-suppressive populations, and promoting cytotoxic T cells infiltration [100,101]. In early-phase clinical trials testing ICIs with chemotherapy in patients with unresectable CCA, ORRs ranged from 36 to 72%, acknowledging differences in study design and dosing regimens [102,103,104].

The following global randomized phase 3 studies TOPAZ-1 and KEYNOTE-966 changed practice establishing combination treatment with anti-PD-(L)1 and chemotherapy as the recommended first-line choice for patients with advanced CCA [15,16]. The two studies had an overall similar design testing standard chemotherapy with cisplatin and gemcitabine combined with an anti-PD-(L)1 agent or placebo followed by a maintenance phase with the anti-PD-(L)1 or placebo alone in the TOPAZ-1 trial and the anti-PD-(L)1 or placebo combined with gemcitabine in the KEYNOTE-966 trial. Both trials met their primary endpoint, showing improved OS in the chemo-immunotherapy arm. Updated results from the TOPAZ-1 trial reported a median OS of 12.0 months (95% CI, 11.6–14.1) versus 11.3 months (95% CI, 10.1–12.5) and a 3-year OS rate of 14.6% (95% CI, 11.0–18.6%) versus 6.9% (95% CI, 4.5–10.0%) with cisplatin plus gemcitabine and the anti-PD-L1 antibody durvalumab compared to chemotherapy-placebo [65]. Similarly, pembrolizumab-chemotherapy achieved longer median OS (12.7 months, 95% CI, 11.5–13.6) versus chemotherapy-placebo (10.9 months, 95% CI, 9.9–11.6) in the KEYNOTE-966 trial, with a maintained benefit after 3 years of follow-up [16,105]. Reassuringly, the performance of chemo-immunotherapy in real-world registries mirrored trial results [106]. To better reflect global treatment patterns and confirm reproducibility in broader patient populations, the efficacy of durvalumab combined with various gemcitabine-based regimens and its use in patients with Eastern Cooperative Oncology Group (ECOG) Performance Status (PS) 2, a group excluded from the registrational trials, is being further investigated in phase 3b studies (NCT05771480, NCT05924880). While validating the role of ICIs in the treatment of CCA, these studies also showed that the added value of these agents was rather modest in an unselected population, fostering the exploration of novel avenues to accomplish the challenging task of targeting the TIME of CCAs.

One such venture has focused on the development of strategies targeting multiple immune checkpoint inhibitors at a time rather than the PD-1/PD-L1 axis alone. Several studies of novel agents targeting PD-(L)1 together with Cytotoxic T-Lymphocyte Antigen 4 (CTLA-4), T cell immunoreceptor with Ig and ITIM domains (TIGIT), or Lymphocyte-activation gene 3 (LAG-3), among others, are ongoing. An example is the phase 2 GEMINI-Hepatobiliary (NCT05775159) trial evaluating volrustomig, an anti-PD-1/CTLA-4 bispecific antibody, or rilvegostomig, an anti-PD-1/TIGIT bispecific antibody, in combination with cisplatin and gemcitabine as front-line treatment in advanced CCA. Of note, rilvegostomig is also being developed in combination with triplet chemotherapy regimens (NCT06569225) or with the antibody-drug conjugate (ADC) AZD8205 targeting B7-H4, a downregulator of T cell function that has been found overexpressed in CCA (NCT05123482). Another negative regulator of the immune response being exploited is lectin-like transcript 1 (LLT1), which reduces NK and cytotoxic T cell activation via CD161 and has been found at high levels in the TIME of renowned immunologically cold tumors like CCAs [107]. A dose-escalation phase 1 trial is studying ZM008, a first-in-class anti-LLT1 antibody, as a single agent followed by combination with pembrolizumab in patients with advanced solid tumors, including CCA (NCT06451497). Boosting the adaptive immune response while orthogonally targeting tumor-associated antigens is a complementary approach. In this space, MDX2001 is a tetraspecific antibody engaging with TROP2 and c-MET on tumor cells and CD3 and CD28 on cytotoxic T cells, improving their activation and proliferation. A phase 1/2a multicenter trial is investigating this molecule in CCA among other solid tumors (NCT06239194). TROP2 and c-MET are also being targeted individually by ADC in early-phase trials open to patients with CCAs (NCT06084481). Additional efforts are developing the use of adoptive T cell therapies. Chimeric antigen receptor (CAR)-T cells targeting CD133 [108] or anti-mucin 1 (MUC1) [109], both highly expressed antigens in CCA, are in preclinical stages, while CAR-T treatments against mesothelin (NCT06256055) and CEA have entered early clinical testing (NCT06043466).

Other paths include targeting tumor angiogenesis, which may represent a strategy to tackle the abundant tumor stroma of CCAs and CAFs, given they largely contribute to VEGF production [87]. While tyrosine-kinase inhibitors (TKIs) [110] and anti-VEGF(R-2) antibodies with [111,112] or without anti-PD-(L)1 [113,114] treatments had proven limited efficacy in an unselected population with refractory CCA, novel studies are evaluating the concomitant dual blockade of VEGF/PD-1 (NCT06529718) and VEGF-A/DLL4 (NCT04492033, NCT05506943) with bispecific antibodies. Both agents are being planned for further study in combination with first-line chemotherapy with (NCT06548412) or without durvalumab (NCT06591520). Lastly, other strategies are testing therapies targeting genomic vulnerabilities deemed to cooperate with the tumor immune escape. Among the known actionable alterations, both *IDH1* mutations and *FGFR2* aberrations have been associated with an immune-suppressive TIME [82] and studies are ongoing to establish if the addition of an anti-PD-(L)1 agent to the *IDH1* inhibitor ivosidenib (NCT06501625, NCT05921760) and the *FGFR2* inhibitor pemigatinib (NCT06530823) could revert this effect and improve outcomes in refractory CCA. An emergent target is *MDM2,* which inactivates *TP53* under physiologic conditions. *MDM2* amplifications have been found in 5–16% of CCAs, particularly in ampullary cancers, and are generally mutually exclusive with the other actionable aberrations [115]. A pooled analysis of two phase 1a/1b trials testing brigimadlin (BI 907828), a highly potent, oral *MDM2*–*TP53* antagonist, with or without the PD-1 inhibitor ezabenlimab and the anti-LAG-3 BI 754111, revealed an encouraging ORR of 50% in patients with refractory *MDM2*-amplified CCA [116]. The subsequent phase 2a/2b Brightline-2 trial (NCT05512377) is ongoing to further study the safety and efficacy of BI 907828 monotherapy in patients with locally advanced or metastatic *MDM2*-amplified, *TP53* wild-type CCA.

Importantly, the establishment of ICIs in advanced CCA has created opportunities to test these agents in earlier disease stages, where outcomes are severely impacted by high relapse rates in patients who undergo curative-intent surgery and by the access to the same limited palliative therapies available for patients with metastatic disease in those with unresectable primary tumors. For patients with resected disease, the multicenter, randomized, double-blind, placebo-controlled phase 3 ARTEMIDE-Biliary01 trial (NCT06109779) is assessing the efficacy of the anti-PD-1/TIGIT bispecific antibody rilvegostomig compared to investigator’s choice of adjuvant chemotherapy. Other ongoing efforts include testing the combination of the anti-PD-L1 tislelizumab and TKI levantinib with capecitabine, the standard backbone adjuvant treatment (NCT05254847). The interim results of this Asian phase 2 study reported a good safety profile and a median disease-free survival (DFS) of 20.23 months (95% CI, 10.84–29.62), while OS is not yet mature [117]. Another open-label, multicenter, phase 2 study (NCT05239169) is currently exploring the association of the anti-CTLA-4 tremelimumab plus durvalumab with or without capecitabine. In the neoadjuvant setting, the phase 2 DEBATE trial (NCT04308174) randomized patients with localized CCA to receive 2:1 neoadjuvant cisplatin and gemcitabine with or without durvalumab, followed by six cycles of durvalumab after surgery. Notably, patients in the chemoimmunotherapy arm achieved almost doubled rates of curative-intent surgery (61% versus 36%) and prolonged progression-free survival (15.1 versus 3.6 months) [118]. Table 1 provides a snapshot of the clinical trials ongoing or planned with immunotherapies in CCA.

## 4. Conclusions and Future Directions

The integration of immunotherapy into the treatment landscape of CCA has marked a significant advancement in a disease with traditionally poor outcomes. The use of ICIs combined with chemotherapy has established a new standard of care for patients with advanced and unresectable CCA, demonstrating OS improvements and opening the door to the exploration of such a strategy in earlier disease settings. Biomarker analyses from early-phase studies revealed associations between response to ICIs and high baseline tumor-infiltrating lymphocytes (TILs) or early on-treatment increase in PD-L1 expression, particularly on immune rather than tumor cells, supporting the concept that immune infiltration is also pivotal for response to immunotherapy in CCA [103,119]. However, baseline PD-L1 expression failed to capture long-term clinical benefits in the larger, registrational-intent phase 3 trials [16,65], making chemoimmunotherapy the currently recommended front-line treatment in all-comers [120]. Given the overall benefit from these combinations remains modest, there is a pressing need to identify robust biomarkers of response and resistance and additional therapeutic targets.

Translational research in CCA has largely suffered from the rarity of the disease and often limited accessibility of tumor lesions, resulting in inadequate quantity and quality of the tissue harvested, particularly from patients with unresectable disease. Thus, most of the studies conducted relied on surgical specimens from primary resected tumors, especially from iCCAs, limiting their generalizability to the broader spectrum of CCAs across sites and stages. While more recent technologies have allowed us to take a more comprehensive look at CCA integrating genomic, transcriptomic, and image analyses, essential work is ongoing to develop tissue-free platforms [121,122,123]. Tumor profiling at this unprecedented resolution led to the identification of distinct molecular phenotypes of CCA beyond conventional anatomic classifications. Additionally, numerous multidimensional CCA datasets with matched genomic and clinical information have been made available on open-source platforms, providing the broader research community with opportunities for further analysis [124]. However, the genomic and immune CCA clusters identified, even those with some utility for patient risk stratification, remain primarily confined to research settings, due to the lack of reproducibility in a routine clinical scenario.

Nonetheless, these studies had the value of showing that druggable genetic alterations are found in a relevant proportion of patients with advanced CCA, allowing the development of targeted treatments and the implementation of tumor profiling as a guideline-recommended clinical practice [120]. Moreover, these studies offered an in-depth characterization of the CCA TIME and demonstrated its crucial role in tumor progression and resistance to therapies, particularly to ICIs. Hosting abundant CAFs, T regs, TAMs, MDSCs, and TANs, among other pro-tumor cell types, the CCA TIME is largely unbalanced toward immune suppression. The intricate, interdependent pro-tumor networks formed by these populations ultimately limit the effectiveness of first-generation immunotherapies. This may also explain why conventional tissue biomarkers of response to ICIs, such as PD-L1 expression, have little to no predictive value in CCA and are therefore not recommended for selecting patients for chemoimmunotherapy in clinical practice. The intrinsically immune tolerogenic liver niche may favor the establishment of a multitude of tumor escape mechanisms beyond the upregulation of the PD-(L)1 axis requiring the study of tumor-specific biomarkers and therapeutic strategies in CCAs.

To address these barriers, drug development efforts are focusing on novel immunomodulatory strategies to enhance cytotoxic immune cell infiltration and activation. Promising approaches involve the simultaneous targeting of multiple immune checkpoints or newly identified molecules with immune-suppressive roles such as LLT1. Others include the synthetical induction of a more proficient anti-tumor immunity, such as with bispecific antibodies rescuing activated cytotoxic T cells or with adoptive cell therapy such as CAR-T cells. Tumor angiogenesis and targetable genomic vulnerabilities are also being exploited as additional drivers of immune evasion with or without immune checkpoint inhibition.

Looking forward, favoring enrolment in clinical trials, establishing international collaborative consortia, and standardizing diagnostic tumor profiling procedures would be key to helping translate basic discoveries into clinical applications. Some of these ambitious programs are the pan-European ENS-CCA and COST Action Precision-BTC-Network which aim to improve the understanding and management of CCA through interdisciplinary collaborations [125,126]. With continued research, the hope is that biomarker-informed and tailored immunotherapies could change the storyline of CCA into an increasingly manageable disease.

## Figures and Tables

**Figure 1 cancers-17-00272-f001:**
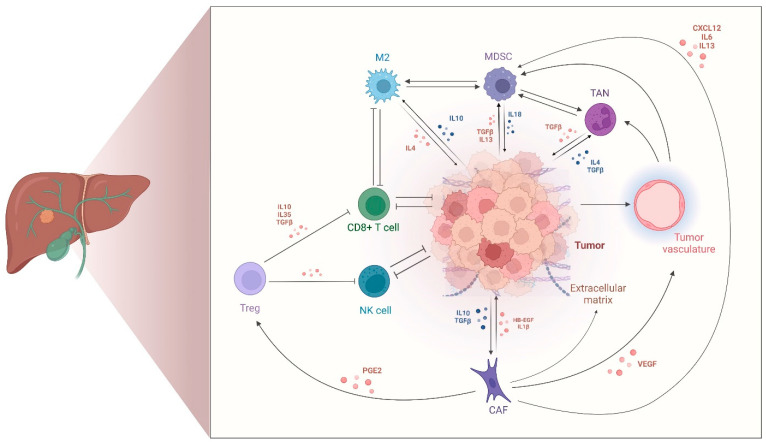
Overview of the main cell types composing the tumor-immune microenvironment of CCA and their functional roles. This figure illustrates the tumor immune ecosystem of CCA with its predominant cell types and their main functional roles. TAMs, MDSCs, TANs, T regs, and CAFs engage in immunosuppressive signaling with tumor and neighboring immune cells, forming a positive feedback loop that drives immune exhaustion, extracellular matrix deposition, neoangiogenesis, and tumor progression. The figure depicts cell–cell interactions, with arrows representing stimulatory signals and T-shaped lines denoting inhibitory signals. CCA, cholangiocarcinoma; TAN, tumor-associated neutrophils; M2, M2-polarized macrophages; MDSC, myeloid-derived suppressor cells; T reg, T regulatory cells; CAF, cancer-associated fibroblasts; CD8+, cytotoxic T cells; NK, natural killers; IL, interleukin; TGFβ, transforming growth factor β; CXCL12, C-X-C motif chemokine 12; VEGF, vascular endothelial growth factor; PGE2, prostaglandin E2.

**Figure 2 cancers-17-00272-f002:**
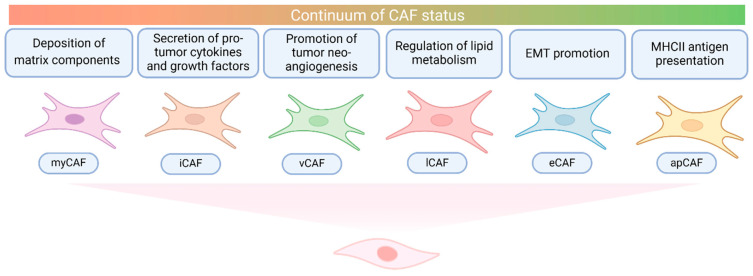
Populations and roles of cancer-associated fibroblasts in cholangiocarcinoma. Main clusters and roles of cancer-associated fibroblasts in cholangiocarcinoma. myCAF, myofibroblastic cancer-associated fibroblast; iCAF, inflammatory cancer-associated fibroblast; vCAF, vascular cancer-associated fibroblast; lCAF, lipofibroblast; eCAF, epithelial to mesenchymal-like cancer-associated fibroblast; apCAF, antigen-presenting cancer-associated fibroblast; EMT, epithelial to mesenchymal transition; MHCII, major histocompatibility complex II.

**Table 1 cancers-17-00272-t001:** Ongoing and planned clinical trials with immunotherapies in CCA.

Study Name/ID	Phase	Status	Experimental Regimen	Target	Class	Setting
**NCT06451497**	1	Recruiting	ZM008 as single agent followed by combination with pembrolizumab	LLT1	Monoclonal antibody	Metastatic
**NCT06256055**	1	Recruiting	UCMYM802	Mesothelin	Cellular therapy (CAR-T)	Metastatic
**NCT06043466**	1	Recruiting	C-13-60	CEA	Cellular therapy (CAR-T)	Metastatic
**NCT06239194**	1/2	Recruiting	MDX2001	CD3/CD28 on T cells and TROP2/c-MET on tumor cells	Tetraspecific T cell engager antibody	Metastatic
**NCT06548412**	1/2	Not yet recruiting	CTX-009 + cisplatin, gemcitabine, durvalumab	VEGF-A/DLL4	Bispecific antibody	Metastatic
**NCT06501625**	1/2	Not yet recruiting	Ivosidenib + cisplatin, gemcitabine, durvalumab	IDH1	Small molecule inhibitor	Metastatic
**NCT05921760**	1/2	Not yet recruiting	Ivosidenib + nivolumab + ipilimumab	IDH1 + PD-1 + CTLA-4	Small molecule inhibitor + monoclonal antibodies	Metastatic
**NCT06708663**	1/2	Not yet recruiting	HX009 + IN10018	PD-1/CD47 + FAK	Bispecific antibody + small molecule inhibitor	Metastatic
**NCT04492033**	1b/2	Active, not recruiting	CTX-009 + irinotecan or paclitaxel	VEGF-A/DLL4	Bispecific antibody	Metastatic
**BLUESTAR** **(NCT05123482)**	1/2a	Recruiting	Rilvegostomig + AZD8205	PD-1/TIGIT + B7-H4	Bispecific antibody + ADC	Metastatic
**GEMINI-Hepatobiliary** **(NCT05775159)**	2	Recruiting	Volrustomig or rilvegostomig + cisplatin and gemcitabine	PD-1/CTLA-4 or PD-1/TIGIT	Bispecific antibody	Metastatic
**NCT06569225**	2	Not yet recruiting	Rilvegostomig + cisplatin, gemcitabine, nab-paclitaxel	PD-1/TIGIT	Bispecific antibody	Metastatic
**SEVILLA** **(NCT06529718)**	2	Not yet recruiting	Ivonescimab	PD-1/VEGF	Bispecific antibody	Metastatic
**NCT06591520**	2	Not yet recruiting	Ivonescimab + cisplatin and gemcitabine	PD-1/VEGF	Bispecific antibody	Metastatic
**NCT06530823**	2	Not yet recruiting	Pemigatinib + durvalumab	FGFR1-3	Small molecule inhibitor + monoclonal antibody	Metastatic
**Brightline-2** **(NCT05512377)**	2	Recruiting	Brigimadlin	MDM2–TP53	Small molecule inhibitor	Metastatic
**NCT06654947**	2	Not yet recruiting	Surufatinib + toripalimab + GEMOX	VEGFR1-3, FGFR1, CSF1R + PD-1	Small molecule inhibitor + Monoclonal antibody	Metastatic
**NCT05506943**	2/3	Active, not recruiting	CTX-009 + paclitaxel	VEGF-A/DLL4	Bispecific antibody	Metastatic
**NCT06591520**	3	Not yet recruiting	Gemcitabine, cisplatin + AK112 or durvalumab	PD-1/VEGF	Bispecific antibody	Metastatic
**TOURMALINE** **(NCT05771480)**	3b	Recruiting	Durvalumab + gemcitabine-based chemotherapy	PD-L1	Monoclonal antibody	Metastatic
**TopDouble** **(NCT05924880)**	3b	Active, not recruiting	Durvalumab + gemcitabine-based chemotherapy	PD-L1	Monoclonal antibody	Metastatic
**DEBATE** **(NCT04308174)**	2	Active, not recruiting	Cisplatin, gemcitabine with or without durvalumab	PD-L1	Monoclonal antibody	Neoadjuvant
**NCT05254847**	2	Recruiting	Tislelizumab + lenvatinib + capecitabine	PD-L1 + VEGFR1-3, FGFR1-4, PDGFRα, KIT, and RET	Monoclonal antibody + multi-TKI	Adjuvant
**NCT05239169**	2	Active, not recruiting	Durvalumab + tremelimumab with or without capecitabine	PD-L1 + CTLA-4	Monoclonal antibodies	Adjuvant
**ARTEMIDE-Biliary01** **(NCT06109779)**	3	Recruiting	Rilvegostomig/placebo + investigator’s choice of chemotherapy	PD-1/TIGIT	Bispecific antibody	Adjuvant

The table provides an overview of the clinical trials ongoing or planned testing immunotherapies for patients with cholangiocarcinoma, as listed on clinicaltrial.gov (accessed on 9 January 2025). Target and class columns summarize drug targets and pharmacological classes for investigational therapies. Abbreviations: LLT1, Lectin-like transcript 1; CAR-T, Chimeric antigen receptor T cells; TROP2, Trophoblastic cell-surface antigen 2; c-MET, Mesenchymal-epithelial transition factor; VEGF(-A), Vascular endothelial growth factor (A); DLL4, Delta-Like Canonical Notch Ligand 4; IDH1, Isocitrate dehydrogenase 1; PD-(L)1, Programmed death-(ligand) 1; CTLA-4, Cytotoxic T-lymphocyte associated protein 4; FAK, focal adhesion kinase; TIGIT, T cell immunoreceptor with Ig and ITIM domains; B7-H4, V-set domain containing T cell activation inhibitor 1; ADC, antibody-drug conjugate; FGFR, fibroblast growth factor receptor; MDM2, Mouse double minute 2 homolog; TP53, Tumor Protein P53; VEGFR, Vascular endothelial growth factor receptor; CSF1R, macrophage colony-stimulating factor 1 receptor; PDGFRα, platelet-derived growth factor receptor-α; KIT, KIT proto-oncogene; and RET, RET proto-oncogene.

## Data Availability

All the data presented within the manuscript is available online through the given references.

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
