# Peer review of "The Immune–Genomics of Cholangiocarcinoma: A Biological Footprint to Develop Novel Immunotherapies"

_cancers, 2025, doi:10.3390/cancers17020272_

Round 1

Reviewer 1 Report

Comments and Suggestions for Authors

In this review, the authors presented a comprehensive overview of the immune biology of cholangiocarcinoma (CCA) and examines how this knowledge has guided clinical drug development, with a focus on both approved and emergent treatment strategies. The use of immune checkpoint inhibitors (ICIs) combined with chemotherapy has established a new standard of care for patients with advanced and unresectable CCA. However, first-generation immunotherapies failed to capture long-term clinical benefit in phase 3 trials. Chemoimmunotherapy remains the currently recommended first line treatment in all-comers. To address these barriers, the authors shed light on the tumor-immune microenvironment (TIME) of CCA and listed the ongoing clinical trials with immunotherapies in CCA. Overall, this is a well-written review. Several concerns with specific points to be considered are listed below:

1.    The metabolic-associated fatty liver disease [MAFLD] in the text is an old terminologies. EASL–EASD–EASO Clinical Practice Guidelines has updated the Nomenclature to metabolic dysfunction-associated steatotic liver disease (MASLD).

2.    In Table 1, it’s better to have another column listing the therapeutic targets (such as the specific genetic mutations) for all the clinical trials.

3.    The font for Line 380-391 needs to be corrected and match the other texts.

4.    It would be interesting to check and mention if there are any public available databases specifically designed for molecular and genomic profiling of CCA.

Reviewer 2 Report

Comments and Suggestions for Authors

Cholangiocarcinoma (CCA) is a highly aggressive and heterogeneous cancer with an increasing global incidence. Genomic and immunological profiling helps to identify molecular subtypes and potentially actionable genetic alterations, which are reviewed in this manuscript. The importance of the tumor immune microenvironment (TIME) in the pathogenesis of CCA and its possible targeting to improve the immune response in hospitalized patients is discussed. In addition, new treatments for CCA are discussed, in particular TIME-targeted therapies and combinations of immune checkpoint inhibitors (ICIs).

The manuscript is well written and is a good contribution to better understanding the biology of CCA and future therapeutic approaches. Although I think it could be published almost as is, I would recommend addressing several minor points.

- There are many abbreviations in the manuscript. For the common reader it would be helpful to have a list of them.

- In Figure 1, a figure legend with a text that guides the reader through the image and provides a few words about the cell types involved could be informative.

- Regarding the “immunological classifications of CCA”, the authors offer different classifications, particularly of iCCA. I wonder if any “conclusions” could be plotted in a figure or pointed out in the text as more reliable.

- A significant and valuable effort has been made to report on ongoing and planned clinical trials with immunotherapies in CCA. Although the compilation was done three months ago, I have found no differences in the reported trials I have reviewed. Could you please take a look to ensure that nothing important has changed and to update it?
